# Soluble Programmed Death Ligand-1 (sPD-L1): A Pool of Circulating Proteins Implicated in Health and Diseases

**DOI:** 10.3390/cancers13123034

**Published:** 2021-06-17

**Authors:** Christian Bailly, Xavier Thuru, Bruno Quesnel

**Affiliations:** 1OncoWitan, Lille, 59290 Wasquehal, France; 2Plasticity and Resistance to Therapies, UMR9020-UMR1277-Canther-Cancer Heterogeneity, CHU Lille, Inserm, CNRS, University of Lille, 59000 Lille, France; xavier.thuru@univ-lille.fr (X.T.); bruno.quesnel@chru-lille.fr (B.Q.)

**Keywords:** immune checkpoint, PD-1/PD-L1, soluble PD-L1, protein maturation, immuno-suppression, cancer, autoimmune diseases

## Abstract

**Simple Summary:**

The interaction of programmed cell death ligand-1 (PDL1) with its receptor PD1 inhibits T-cell responses. Blockade of this interaction with monoclonal antibodies leads to major antitumor effects. However, not all cancer patients respond well to anti-(PD-1/PD-L1) immunotherapy. The PD-L1 protein is expressed at the cell plasma membrane (mPD-L1), at the surface of exosomes (exoPD-L1), in cell nuclei (nPD-L1) and as a soluble circulating protein (sPD-L1). The aim of our analysis was to highlight the multiple variants of sPD-L1 generated either by the proteolytic cleavage of m/exoPD-L1 or by the alternative splicing of PD-L1 pre-mRNA. The objective was also to underline the presence and role of circulating sPD-L1 isoforms in multiple cancer indications and many other diseases (including chronic inflammatory and viral diseases), and under non-pathological conditions (pregnancy). sPD-L1 often represents a general marker of an inflammatory status. The pool of sPD-L1 proteins is an integral part of the highly dynamic PD-1/PD-L1 signaling pathway.

**Abstract:**

Upon T-cell receptor stimulation, the Programmed cell Death-1 receptor (PD-1) expressed on T-cells can interact with its ligand PD-L1 expressed at the surface of cancer cells or antigen-presenting cells. Monoclonal antibodies targeting PD-1 or PD-L1 are routinely used for the treatment of cancers, but their clinical efficacy varies largely across the variety of tumor types. A part of the variability is linked to the existence of several forms of PD-L1, either expressed on the plasma membrane (mPD-L1), at the surface of secreted cellular exosomes (exoPD-L1), in cell nuclei (nPD-L1), or as a circulating, soluble protein (sPD-L1). Here, we have reviewed the different origins and roles of sPD-L1 in humans to highlight the biochemical and functional heterogeneity of the soluble protein. sPD-L1 isoforms can be generated essentially by two non-exclusive processes: (i) proteolysis of m/exoPD-L1 by metalloproteases, such as metalloproteinases (MMP) and A disintegrin and metalloproteases (ADAM), which are capable of shedding membrane PD-L1 to release an active soluble form, and (ii) the alternative splicing of PD-L1 pre-mRNA, leading in some cases to the release of sPD-L1 protein isoforms lacking the transmembrane domain. The expression and secretion of sPD-L1 have been observed in a large variety of pathologies, well beyond cancer, notably in different pulmonary diseases, chronic inflammatory and autoimmune disorders, and viral diseases. The expression and role of sPD-L1 during pregnancy are also evoked. The structural heterogeneity of sPD-L1 proteins, and associated functional/cellular plurality, should be kept in mind when considering sPD-L1 as a biomarker or as a drug target. The membrane, exosomal and soluble forms of PD-L1 are all integral parts of the highly dynamic PD-1/PD-L1 signaling pathway, essential for immune-tolerance or immune-escape.

## 1. The PD-1/PD-L1 Checkpoint

The inhibitory checkpoint molecule programmed death-1 (PD-1) plays a vital role in maintaining immune homeostasis upon binding to its ligands PD-L1 and PD-L2. PD-1 (CD279) is an inhibitory receptor induced in activated T-cells and its two ligands are necessary to maintain peripheral tolerance. In addition, PD-L1 (CD274, B7-H1) plays a key role as a negative regulator of antitumor immunity, and the blockade of the PD-1/PD-L1 interaction permits one to restore and to increase the function of T-cells. Monoclonal antibodies directed against PD-1 (Nivolumab, Pembrolizumab, Cemiplimab, Spartalizumab, Camrelizumab, Sintilimab) or PD-L1 (Atezolizumab, Durvalumab, Avelumab, SHR-1316) are now used regularly for the treatment of multiple types of solid tumors, such as non-small cell lung cancer (NSCLC), melanoma, and gastric cancers to cite only a few types [1]. Since the first approval of pembrolizumab in 2014 for the treatment of advanced or unresectable melanoma, seven antibodies targeting PD-1 or PD-L1 have been approved by the Food and Drug Administration (FDA) for the treatment of solid tumors or onco-hematological diseases (lymphoma) [2]. These immune checkpoint inhibitors (ICI) have greatly improved patient survival in many advanced malignancies, but not all. Clinical responses are extremely variable from one cancer to another, and the durability of the response is also very variable from one patient to another. Spectacular responses have been observed in NSCLC and melanoma, whereas the response rate is much more limited in neuro-oncology, for example [3]. Similarly, PD-1/PD-L1 checkpoint inhibitors are not very effective in treating acute myeloid leukemia [4]. Only a small fraction of cancer patients respond well to PD-1/PD-L1 blockade. Combinations of chemo- or radiotherapy with immuno-therapy are largely developed to improve cancer treatment [5,6].

### 1.1. Membrane PD-L1

PD-1 and PD-L1 are both membrane proteins expressed on immune cells and cancer cells, respectively. The success of an anti-PD-1/PD-L1 therapy greatly depends on the capacity of the injected antibody to activate T-cells to eliminate tumor cells, with the support of tumor-infiltrating lymphocytes. The membrane expression of PD-1 and PD-L1 is crucial to cancer immunotherapy, although in some cases there is little or no significant difference in overall survival between patients with PD-L1-positive and PD-L1-negative tumors [7]. Expression of this immune checkpoints is not limited to T-cells and cancer cells. PD-1 is primarily expressed on the surface of T-cells, including regulatory T-cells (Treg), B cells, monocytes, dendritic cells (DC), and natural killer (NK) cells. Moreover, PD-1 expression can be induced by different cytokines (such as TGF-β and interleukin-10) on antigen-presenting cells (APCs), myeloid DC and monocytes, and under different pathological conditions such as preeclampsia, chronic viral infections, and cancer [8]. On the other hand, PD-L1 is often highly expressed on tumor cells and can be present on activated T and B cells, DCs, monocytes and occasionally on endothelial and epithelial cells. Its expression can be induced by different inflammatory cytokines and interferon-γ (IFN-γ). The PD-1/PD-L1 checkpoint is associated with inflammatory effects, and it is now well established that the checkpoint plays a role in multiple diseases and conditions beyond cancer [9]. Notably, the PD-1/PD-L1 checkpoint is largely implicated in reproductive immunology, with an important role in the establishment of immune tolerance mechanisms at the materno–fetal interface [10].

In the oncology field, numerous studies have defined the potential values of PD-1/PD-L1 membrane expression as biomarkers for immunotherapy effectivity and tumor resistance in multiple cancer types. In general, a prominent PD-L1 expression on tumor cells and high levels of activated T-cells have been associated with a higher response rate. There is a satisfactory correlation between PD-L1 expression on tumoral cells and patients’ response to immune checkpoint inhibitors. The level of PD-L1 expression on circulating tumor cells (CTCs) is also considered a favorable biomarker in patients treated with ICIs, at least in the cases of NSCLC and melanoma [11,12,13]. PD-L1 expressed on CTCs is also viewed as a useful biomarker for the early detection of cancers [14]. However, it requires specific methods for CTC enrichment and the immune detection of PD-L1^+^ CTC, using harmonized procedures [15].

### 1.2. Exosomal PD-L1

Besides the main form of PD-L1 expressed at the cell surface, notably on cancer cells, there are two other important forms of PD-L1 with distinctive roles (Figure 1). PD-L1 can be expressed on cancer-derived exosomes, which are biologically active extracellular vesicles released from different types of tumor cells. These vesicles, with a cell membrane-like lipid-bilayer, can recapitulate the effect of cell-surface PD-L1 and they can significantly modulate the response to anti-PD-1/PD-L1 antibody therapy. They play a role in intercellular communications, in the composition and dynamic of the extracellular environment, and they can significantly affect the reactivity of the immune system [16]. Exosomal PD-L1 (exoPD-L1) can be considered a mimic of tumor cell membrane PD-L1, capable of inhibiting T-cell immunity and enhancing the growth of various tumor types. Exosomal PD-L1 directly contributes to immunosuppression, not only in cancer [17] but also in wound healing [18]. There are several recent, comprehensive reviews about the biology and specific roles of exoPD-L1 [19,20,21,22,23,24]. Exosomal PD-L1 is a target similar to cell membrane PD-L1 (mPD-L1) and different approaches have been designed to reduce exosome biogenesis, secretion or to neutralize secreted exosomes, as recently discussed by Yin and coworkers [25].

### 1.3. Soluble PD-L1

A third important PD-L1 entity is the soluble form, not anchored into a plasma membrane or a vesicle, but free in solution (Figure 1). It is a circulating form, detected in the serum of cancer patients [26], patients with auto-immune diseases or some viral diseases [27,28], and other pathologies and conditions, including in pregnant women [29]. sPD-L1 has been identified in more than 20 different pathologies and often plays an important immuno-regulatory role (Figure 2). In cancer, both soluble forms of the checkpoint, soluble PD-1 (sPD-1) and soluble PD-L1 (sPD-L1), have been detected in plasma, and elevated levels have been associated with advanced disease and worst prognosis for patients [30]. There are different methods to detect and measure the level of sPD-L1 in the serum and plasma [31,32,33,34]. Its role as a prognostic or predictive marker in lung cancer has been debated recently but the significance of this soluble protein remains uncertain [35]. Here, we provide an update on the origin, biology and potential roles of sPD-L1 in different pathologies.

Our analysis deals with the soluble form of PD-L1, distinct from the exosomal form exoPD-L1, which is a membrane-bound form similar to mPD-L1. This point is essential because it requires robust methods to distinguish the free, soluble protein from the protein associated with extracellular vesicles (EV). Purification kits commonly used to isolate EV from biofluids, such as plasma or serum, may not always totally eliminate the free protein. Conversely, analysis of sPD-L1 in circulation using immune-assays (ELISA) may not distinguish between vesicular and soluble forms. This is a key aspect, perhaps insufficiently considered in some studies, although there are specific approaches to distinguish the two forms [36,37].

The intracellular form of PD-L1 shall also not be neglected. It is a bioactive form, notably acting as an RNA-binding protein to regulate the mRNA stability [38]. Nuclear PD-L1 is believed to play an important role in cancer cells independent of its function in immune checkpoint [39,40]. However, for the sake of clarity, we will not discuss further the PD-L1 intracellular species here.

**Figure 2 cancers-13-03034-f002:**
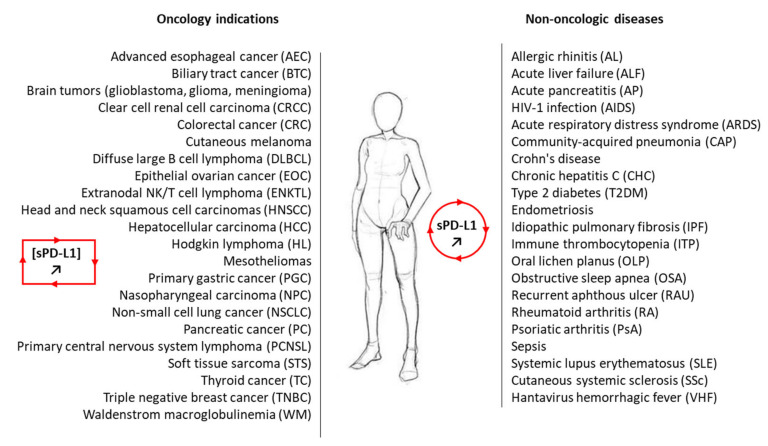
A non-exhaustive list of malignant and non-malignant diseases for which sPD-L1 has been measured in the plasma of patients. In most cases, the level of circulating sPD-L1 was found to be enhanced in the plasma of patients with the indicated pathology compared to healthy control (see Table 1. For the oncology indications, the presence of sPD-L1 and its significance are discussed in the following studies: AEC [41]; BTC [42,43]; brain tumors [26,44]; CRCC [45,46]; CRC [47]; cutaneous melanoma [48]; DLBCL [49,50]; EOC [51,52]; ENKTL [53,54]; HNSCC [36]; HCC [55,56,57]; HL [54,58]; mesothelioma [59,60]; NC [61,62]; PGC [63,64,65]; NSCLC [66,67]; PC [68]; PCNSL [50]; STS [69,70]; TC [71]; TNBC [72]; WM [73].

## 2. Generation of Soluble PD-L1: Proteolysis and Alternative Splicing

### 2.1. Structure and Domain Organization of mPD-L1/exPD-L1 Protein

The PD-L1 immune checkpoint molecule belongs to the B7 family of ligands, together with PD-L2, B7-H3, HHLA2, and B7x [74]. Its protein structure is similar to other members of the family with an N-terminus peptide, two immunoglobulin-like extracellular domains IgV (amino acids 19–127) and IgC (amino acids 133–225), with four specific *N*-glycosylation, three phosphorylation sites, one palmitoylation site on the intracellular domain (C272), and one ubiquitination site (K178) (Figure 3). PD-L1 is subject to intense post-translational regulations [75]. The IgV domain is the region responsible for the binding function of PD-L1 and represents the target site for monoclonal antibodies, peptides and small molecules [76]. A short stalk region connects the IgC domain to the transmembrane domain. This flexible segment, between amino acid residues P227 and R238, is important because it contains the proteolytic cleavage sites (discussed below). It includes an 11-amino acid sequence, including four proline residues, considered important for the interaction in cis with the protein partner B7-1 which can competitively block the binding of PD-L1 to PD-1. Indeed, PD-L1 binds to PD-1 but also binds to B7-1 (CD80) to regulate T-cell function [77,78]. The sequence of the stalk segment is variable from one species to another, being distinct between human, dog, chicken, rat and mouse, for example (Figure 4). These differences may have significant consequences because, as pointed out here, this region is not inert but seems to play a role in the proteolytic processing of the protein. PD-L1 inhibitors can inhibit the PD-L1/B7-1 interaction [79].

PD-L1 is often expressed on tumor cells but also on tumor-associated fibroblasts, which play a major role in the control and maintenance of the tumor microenvironment. These fibroblasts can express PD-L1 at their membrane surface, and the expression can be induced by transforming growth factor-beta (TGF-β). The interplay between PD-L1 and TGF-β, a potent profibrotic cytokine, is essential to control fibroproliferation. The stimulation of fibroblasts with TGF-β leads to the enhanced secretion of extracellular vesicles containing PD-L1 which were found to decrease T-cell proliferation and to increase fibroblast migration in a PD-L1-dependent manner [80]. In this case, exoPD-L1 exerted a clear immunosuppressive action; a picture entirely coherent with the inhibition of cytokine production by CD^8+^ T-cells and their suppression observed with exPD-L1 in other situations [18]. Via the expression of PD-L1, cancer-associated fibroblasts have a major immunosuppressive action on tumor cells [81] but the immunosuppression can be lifted or reduced by targeting PD-L1 directly or indirectly via TGF-β, for example. Similarly, PD-L1 expressed on invasive fibroblasts plays a role in fibrosis [82]. Targeting fibroblast-associated PD-L1 is considered an option to combat idiopathic pulmonary fibrosis [83,84,85].

**Figure 4 cancers-13-03034-f004:**
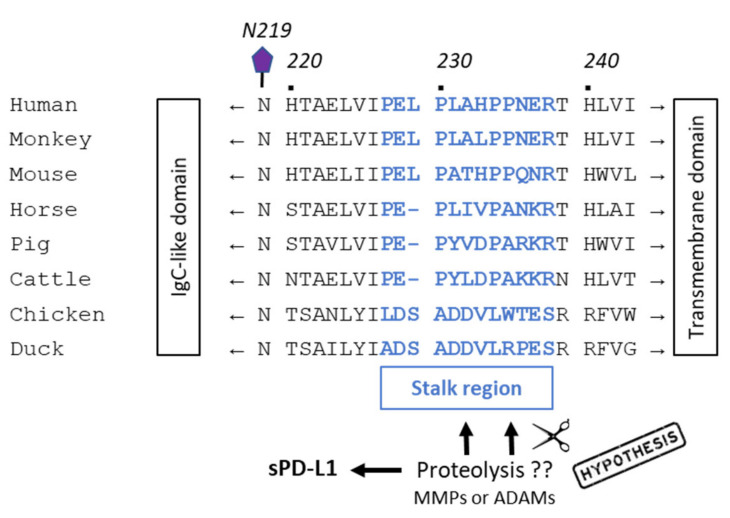
Alignment of amino acid sequences of PD-L1 from humans with other species. A stalk region of 10–12 amino acids (in bold blue) is located between the IgC-like domain and the transmembrane domain. This region would correspond to the site of cleavage of mPD-L1 by proteases, MMPs and/or ADAMs, to release sPD-L1 [83]. Amino acid sequences were derived from [86,87].

### 2.2. Proteolytic Generation of sPD-L1

The release of PD-L1-containing vesicles from fibroblasts is a mechanism to enhance immunosuppression. Conversely, the decreased expression of mPD-L1 from the surface of fibroblasts is a mechanism to weaken immunosuppression. The expression of mPD-L1 can be controlled at the transcriptional level, but also at the protein level when the ligand is inserted into the plasma membrane via the action of specific proteases. It has been demonstrated that myofibroblasts expressing PD-L1, such as those in the inflamed mucosa of patients with Crohn’s disease, exhibit an increased expression of matrix metalloproteinases (notably MMP-10) and a concomitant decreased expression of mPD-L1. The reduction in mPD-L1 level on myofibroblasts impairs their capacity to control T helper cells [26]. Different studies have demonstrated that mPD-L1 expression can be regulated by MMP-mediated proteolytic cleavage. mPD-L1 expressed on infant foreskin fibroblasts can be selectively cleaved by recombinant human MMP-9 and -13, two secreted MMPs, and this action abrogates T-cell apoptosis induced by the fibroblasts. The MMP cleavage removes the PD-1 binding domain of mPD-L1 [88]. Notably, mPD-L1 on fibroblasts can be selectively cleaved by MMP-13 and this process would limit their immunosuppressive capacity and would exacerbate the inflammatory status in tissues. Both PD-1 and PD-L2 are sensitive to MMP cleavage. PD-L2 appeared sensitive to a broad range of MMP activities, whereas PD-L1 was found to be selectively cleaved by MMP-13. This work was the first demonstration of a proteolytic generation of sPD-L1 from mPD-L1 [88]. The cleavage or shedding of mPD-L1 by MMP-13 was further evidenced using human oral squamous cell carcinoma (OSC-19 and OSC-20 cell lines, both strongly expressing mPD-L1). In this case, it was shown that both MMP-7 and MMP-13 were able to degrade PD-L1, either the full-length protein (56-kDa) or its C-terminal truncated form (46-kDa). The use of specific inhibitors of MMP-13 could restore mPD-L1 expression in these cells [89]. Altogether, the three studies evidence a proteolytic origin of sPD-L1, from membrane-anchored PD-L1 after cleavage or shedding by different MMPs, notably MMP-13. However, more work is needed to better determine the origin of sPD-L1 in disease-specific contexts.

Another group of cell surface metalloproteases, the ADAMs (A disintegrin and metalloproteases), is largely implicated in the shedding of protein ectodomain and paracrine signal transduction. Notably, ADAM10 and ADAM17 (TACE) play roles in gastroenterological tumors, contributing to different processes, such as a reduction in DNA damage repair, tumor growth, vascularization, and in the control of inflammatory responses in the intestine [90,91,92]. These two membrane-bound ADAM proteases can mediate the cleavage of mPD-L1 and release sPD-L1, as a 37-kDa N-terminal PD-L1 fragment, in breast cancer cell lines. The C-terminal fragment (18-kDa), including the PD-L1 cytoplasmic domain, which remains associated with cells, is not stable, and rapidly eliminated by lysosomal degradation. The release of sPD-L1 to the media can be significantly reduced with the use of siRNAs targeting both ADAM10/17 or with specific small molecule ADAM10/17 inhibitors, but not with MMP-9/13 inhibitors [93]. The shedding of PD-L1 was not only observed in breast cancer cells (MDA-MB-231, MCF10A), but also with other cell lines derived from prostate (DU-145) or lung (A549) cancers. The position of the mPD-L1 cleavage site by ADAMs is not precisely known but the authors postulated that it is located in the stalk region, between V225 and H240, based on the analysis of the size of the degradation products (Figure 4) [93]. This is an interesting hypothesis, but it requires an experimental validation. A recent study has revealed that both ADAM10 and ADAM17 can cleave PD-L1 from the surface of malignant cells (such as 786-0 renal cell carcinoma that express both the checkpoint ligand and the two proteases) but also extracellular vesicles. ADAM proteases are considered cleavers of PD-L1 from the surface of cancer cells (e.g., bladder tumor cells) and possibly also from immune cells [94]. ADAMs, ADAM10 in particular, are known to be present in small extracellular vesicles and responsible for ectodomain shedding [95,96]. Both ADAM10 and ADAM17 can cleave PD-L1 from the surface of vesicles and cells [97]. ADAM10 has been shown to release different proteins, such as the transmembrane receptor CD30 implicated in Hodgkin Lymphoma, releasing it either as a molecule embedded in the membrane of extracellular vesicles or as a cleaved soluble ectodomain [98]. The protease does the same thing with PD-L1 and there are probably other “sheddases” contributing to the release of sPD-L1 from cells. The proteolytic removal of membrane protein ectodomains is a common mechanism for many receptors and implicates numerous proteases [99].

### 2.3. Alternative Splicing Generation of sPD-L1

sPD-L1 can arise from a non-proteolytic process, via an alternative splicing. The PD-L1 gene (and corresponding PD-L1 pre-mRNA) can generate many different splice variants, including a long non-coding RNA isoform (PD-L1-lnc) [100,101] and a variety of variants leading to truncated proteins. These isoforms have been shown to play distinct roles in the regulation of immune surveillance and progression in colorectal cancer [102,103]. sPD-L1 can be produced by the acquisition (the so-called exaptation process) of an intronic element (LINE-2A) in the PD-L1 gene, which causes the omission of the transmembrane domain and the associated regulatory sequence. The resulting alternatively spliced transcript encodes an sPD-L1 form devoid of T-cell inhibitory activity [104]. Similarly, a splice variant lacking the transmembrane domain produces a secreted form of PD-L1 with a specific tail domain capable of homodimerization, as represented in Figure 5 [105]. Other secreted PD-L1 splicing variants (without transmembrane domain) have been characterized, including in patients with non-small cell lung cancer [106], and in melanoma [48] and leukemia cells [102,107]. There are different splice forms of PD-L1 (such as exon 4-enriched variants) capable of generating a secreted form of PD-L1. They are present in numerous cancer types [108]. The soluble variants of these receptors contribute to immune regulation [109]. However, as reported below, their functions can be quite distinct from one form to another.

To sum up this section, sPD-L1 can be generated essentially via two mechanisms, either from a proteolytic cleavage of the (plasma or vesicular) membrane-bound form or from an alternative splicing of PD-L1 pre-mRNA which generates mRNAs and then soluble isoforms lacking the transmembrane domain.

**Figure 5 cancers-13-03034-f005:**
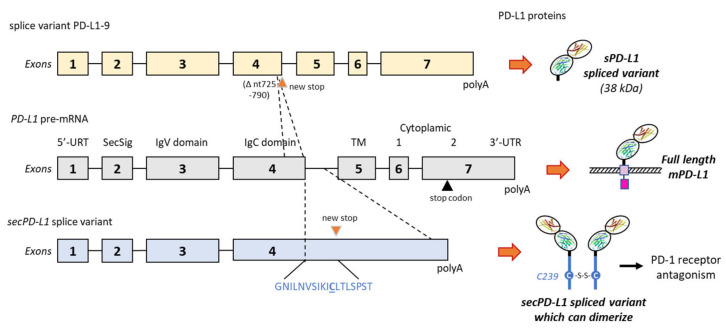
Two examples of alternative splicing of the PD-L1 pre-mRNA. The exons’ and introns’ organization of full-length PD-L1 is shown in the middle, and the full-length PD-L1 protein is shown on the right with its different domains (as in Figure 1). Above, the splice variant PD-L1-9, which has lost a 66-bp region from nt-725 to 790 in exon 4. The deletion indicates a frame shift leading to a stop codon before the transmembrane domain (TM). The variant produces a truncated sPD-L1 protein (38 kDa) lacking the TM and intracellular domains [48]. Below is an alternatively spliced form of the human PD-L1 cDNA from placental tissue. The variant contains the first 4 exons of PD-L1, including the secretory signal (SecSig) at the N-terminus, IgV and IgC domains, which are shared with the full-length PD-L1. The variant does not splice into the fifth exon (encoding the transmembrane domain) but reads into the fourth intron, within a new stop codon. It produces an mRNA that lacks a transmembrane domain at its 3′ end and leads to a protein with the indicated unique C-terminal sequence. The underlined cysteine residue (C239) allows for protein homodimerization, as represented on the right side. The expressed protein, naturally dimerizing, was found to inhibit T-cell proliferation and production of IFN-γ from activated T-cells [105].

## 3. Significance of Soluble PD-L1 in Cancer

### 3.1. sPD-L1 as a Cancer Biomarker

Circulating soluble PD-L1 has been largely exploited as a diagnostic, therapeutic, or prognostic biomarker for cancers. The presence of sPD-L1, generated by the proteolytic cleavage of mPD-L1 and/or by the translation of alternatively spliced PD-L1 mRNA, has been confirmed in many types of cancers, such as those listed in Figure 2 [110]. The molecule can be easily detected in the serum and plasma by various methods (mostly immuno assays (ELISA)) and its presence at a high level generally predicts a better response to an anti-PD-(L)1 mAb. More expression of sPD-L1 makes the response to anti-PD1 therapy more likely, in many cases (but not always). At the same time, a high level of soluble PD-L1 in peripheral blood often predicts poor prognosis in patients with solid tumors [111]. However, as an isolated marker, it cannot be considered a predictor of overall survival of patients with solid tumors [63]. The level of sPD-L1 does not always correlate with response to immune-sensitivity, notably in lung cancers [45,66]. In some cases, the level of sPD-L1 is higher for metastatic patients compared to non-metastatic patients, as observed in clear cell renal cell carcinoma [45], but again there is no general rule. The circulating levels of sPD-L1 represent a predictive biomarker of clinical response to anti-PD-L1 in mesothelioma patients [59], and it might be a useful marker to predict the outcome in glioma patients receiving radiotherapy [112]. Measurements of sPD-L1 could be useful to predict metastasis and prognosis in soft tissue sarcoma and hepatocellular carcinoma [55,69]. An elevated serum level of sPD-L1 may also represent adverse prognostic factor in certain subtypes of T-cell lymphomas and leukemias [113,114], but there is no general rule for all cancers. sPD-L1 levels vary enormously according to the lymphoma type. sPD-L1 levels are higher in patients with diffuse large B-cell lymphoma compared to healthy individuals but lower in patients with follicular lymphoma [114]. In most of these studies, sPD-L1 is detected by ELISA, but the exact nature of the soluble protein (long form or truncated variants) is not determined. A better characterization of the circulating sPD-L1 form is recommended to better appreciate the potential role. We will not comment further on the biomarker aspect of sPD-L1, as there are specific recent reviews on this topic [115,116,117,118,119,120,121].

The level of circulating sPD-L1 measured in the plasma or serum can vary considerably from one tumor type to another, and it can depend also on the stage of the disease, gender, and the associated treatment. Even in a defined cancer type and with a comparable method (ELISA), large differences of sPD-L1 levels have been reported, complicating the comparisons. For examples, in locally advanced non-small cell lung cancer (NSCLC), a median serum sPD-L1 concentration of 67–68 pg/mL has been reported [66,122], whereas other studies indicated median sPD-L1 levels of 27 pg/mL [123], 84 pg/mL [124] and 176 pg/mL [125]. Moreover, much higher values have been reported with other methods, up to 568 pg/mL measured using a multiplex assay [67] and 3.84 ng/mL measured with another ELISA procedure [126]. A quantitative comparison is thus difficult, if not impossible. Moreover, these measurements refer to the presence of sPD-L1 in the serum of patients, not the functionality of the circulating protein.

### 3.2. Functionality of sPD-L1 in Cancer

The question of the functionality of sPD-L1 in cancer is still disputed. Does sPD-L1 bind to PD-1? Yes. Does sPD-L1 deliver a positive or a negative regulatory signal through PD-1? The question remains open at present, although in most cases the answer to the question is “negative regulatory signal”. Different studies have indicated that sPD-L1 is a functional, glycosylated protein, capable of binding to PD-1 [127]. There are engineered variants of sPD-L1 (generated with directed molecular evolution) with an affinity over 20 folds greater than that of native human PD-L1 and able to compete with an anti-PD-1 antibody [101]. In this study, native human sPD-L1 was found to induce suppressive effects on activated T-cells, thus mimicking the effect of mPD-L1 [128]. A recent study using sPD-L1 isolated from the plasma of patients with recurrent/metastatic breast cancer demonstrated that sPD-L1 can inhibit T lymphocyte function, acting as a negative regulatory element in cellular immunity [129]. Similar observations have been reported previously in experimental (in vitro) studies using sPD-L1 released from lung cancer cells [130]. sPD-L1 had been found to exert an immuno suppressive role either in inhibiting T-cell activation or promoting T-cell apoptosis [131].

Similarly, sPD-1 also represents an active circulating protein, with an immune-modulatory capacity [132]. Its activity can be exploited to combat cancer. Designed chimeric antigen receptor (CAR) T-cells which constitutively secrete sPD-1 have shown a marked antitumor activity [133]. Proteolytic sPD-L1, notably the form released after the ADAM-cleavage of mPD-L1, is an active circulating protein capable of inducing apoptosis in CD^8+^ T-cells and compromising the killing of tumor cells by these effector cells [97]. Consequently, because it is an active signaling molecule, sPD-L1 can significantly influence the efficacy of therapeutic antibodies targeting the checkpoint. sPD-L1 could play the role of a sink for the injected PD-L1 inhibitors (antibodies or small molecules), acting as a decoy receptor for anti-PD-L1 antibodies, in the frame of a resistance mechanism [106]. As a corollary, the therapeutic anti-PD-L1 entity can be useful to target simultaneously the different active species that are mPD-L1, exPD-L1 and sPD-L1. In particular, the targeting of sPD-L1 could be interesting to reinforce the antitumor effect of existing therapy (see below).

In most cases, sPD-L1 has been shown to function as a PD-1 blocker, as its parent product mPD-L1. However, a distinct situation has been reported with an sPD-L1 variant issued from alternative splicing, the above-mentioned form produced by exaptation of a LINE-2A intronic element. In this case, the specific sPD-L1 form lacked measurable T-cell inhibitory activity and, in sharp contrast, it was found to function as a PD-1 receptor antagonist, blocking the inhibitory activity of its competitors mPD-L1 and exPD-L1 [104]. In contrast, another study pointed out a specific splice variant producing a form of sPD-L1 (42-kDa) without the transmembrane domain but with a unique 18-amino acid tail (GNILNVSIKICLTLSPST) bearing a key cysteine (C) residue which allows the protein to homodimerize (Figure 5). In this case, the splice form sPD-L1 was found to inhibit T-cell proliferation (without cell-to-cell interaction) and production of IFN-γ. It is important to underline that this form of sPD-L1 can be produced by many types of cancer cells (almost always in addition to mPD-L1) but also in normal cells [105]. This specific sPD-L1 form is not exclusive to tumor cells, corroborating the detection of sPD-L1 in multiple other pathologies and conditions (see below). Different types of PD-L1 splice variants can be generated depending on the cells used and experimental conditions. For example, in melanoma cells (A375 and M34), up to four variants have been identified in addition to full-length PD-L1, with splices occurring from the exon 4 to exon 6, thus leading to the expression of various forms of sPD-L1. Three secreted forms of sPD-L1 (24, 38, and 45-kDa) have been detected in the supernatants of cultured melanoma cells. They were likely issued from the alternative splicing of the PD-L1 transcript, and they all have inhibitory functions on T-cell activation and proliferation [48]. For example, the variant PD-L1-9 (Figure 5) has lost a 66-bp region in exon 4, inducing a frame shift leading to a stop codon before the transmembrane domain. In melanoma patients, three major splice variants of sPD-L1 originating from both tumor and immune cells, and differentially secreted, have been characterized. Their respective immunosuppressive role remains to be clarified [48].

Another immuno-active form of sPD-L1 has been identified in a large subset of cancer cell lines that express a high level of the PD-L1 gene with an exon-4 enrichment. Many cell types with a high expression level of exon 4 (e.g., RKO colon cancer cells; CAL62 thyroid anaplastic carcinoma cells) were found to express a truncated PD-L1 isoform which retained the ability to bind PD-1 and to function as a negative regulator of T-cell function, inhibiting IL-2 and IFN-γ secretion in primary T-cells [108].

Therefore, according to the splicing process, an immuno-suppressive or a non-immuno-suppressive variant of sPD-L1 can be generated. The immuno-suppressive status may depend on the capacity of the secreted protein to dimerize, because PD-L1 dimerization is required to inhibit the activation of T lymphocytes. However, this hypothesis, entirely possible, remains to be validated. It is clear that there are multiple forms of circulating sPD-L1, cleaved forms generated by proteases and different alternative splicing variants. Altogether, they provide a structurally and functionally variable pool of soluble PD-L1 species.

## 4. Soluble PD-L1 beyond Cancer

It has been reported recently that sPD-L1 could be considered a marker of both local and systemic inflammation in patients with a brain tumor, such as glioblastoma and glioma [44]. The link between sPD-L1 and general inflammation has been underlined in other pathologies, well beyond cancer. sPD-L1 has been detected in various pathologies, often associated with markers of inflammation. These non-oncology pathologies are extremely diverse, including idiopathic pulmonary fibrosis, acute pancreatitis, viral infections such as HIV-infection and several others (Table 1). Most of them are diseases mediated by T-cells. These pathologies are briefly evoked here to underline the link with sPD-L1 (and/or sPD-1).


cancers-13-03034-t001_Table 1Table 1Expression status of sPD-L1 detected in non-oncology pathologies.Pathology or ConditionsPD-L1 Status and/or FunctionReferencesAllergic rhinitis (AL)Increased expression levels of both sPD-1 and sPD-L1 in peripheral blood of AR patients.[134]Acute liver failure (ALF)sPD-L1 plasma levels increased in patients with ALF, notably in patients who developed sepsis or had a poor outcome.[135]Acute pancreatitis (AP)Higher serum sPD-1 levels in AP patients with infection complications compared to patients without complication. Upregulation of sPD-L1 in patients with early AP and infectious complications.[136,137]HIV-1 infection (AIDS)Much higher levels of sPD-L1 in HIV-infected patients compared to uninfected adults.[138,139]Acute respiratory distress syndrome (ARDS)sPD-L1 upregulated in survivors of direct ARDS compared to non-survivors. sPD-L1 induces apoptosis of monocyte-derived macrophages in ARDS patients.[140]Community-acquired pneumonia (CAP)Higher level of circulation sPD-L1 in patients with severe CAP compared to CAP group and healthy controls. Correlation between sPD-L1 level in CAP patients and survival prognosis.[141]Crohn’s diseasemPD-L1 cleaved from the cell surface by MMP-10 to generate a soluble form of PD-L1.[28]Chronic hepatitis C (CHC)High level of serum sPD-L1 in CHC patients associated with disease progression.[142]Type 2 diabetes (T2DM)Elevated amount of sPD-L1 (and IFN-g) in the sera of patients with T2DM compared to controls, notably in T2DM patients with an acute coronary syndrome.[143]EndometriosisElevated level of sPD-L1 in the serum and peritoneal fluid of patients with endometriosis vs. control.[144]Idiopathic pulmonary fibrosis (IPF)Elevated concentrations of sPD-L1 in the serum of IPF patients compared to healthy population.[27]Immune thrombocytopenia (ITP)Decreased levels of sPD-L1 in patients with newly diagnosed ITP compared to patients with chronic ITP.[145]Oral lichen planus (OLP)Higher expression of sPD-1 and sPD-L1 in patients with OLP than in control group. Negative correlation between sPD-L1 expression level and CD^4+^ T lymphocytes.[146]Obstructive sleep apnea (OSA)higher levels of sPD-L1 in severe OSA compared to mild OSA or non-OSA patients.[147,148]Recurrent aphthous ulcer (RAU)Higher levels of both sPD-1 and sPD-L1 in RAU patients compared to control group.[149]Rheumatoid arthritis (RA) Psoriatic arthritis (PsA)Increased concentrations of sPD-1 and sPD-L1 in knee synovial fluid and serum in the rabbits of the RA-model group compared to control. Increased levels of sPD-1 in RA and PsA.[150,151]SepsisHigh levels of circulating sPD-L1 in sepsis, positively correlated with the sepsis severity.[135,152]Systemic lupus erythematosus (SLE)Higher levels of both sPD-1 and sPD-L1 in SLE patients compared to control group.[153]Cutaneous systemic sclerosis (SSc)Elevated levels of sPD-L1 in patients with diffuse or limited cutaneous SSc. Possible marker of the severity of skin sclerosis.[154]Hantavirus-associated virus hemorrhagic fever (VHF)High amounts of sPD-L1 and sPD-L2 in sera from hantavirus-infected patients.[155]


The measured circulating level of sPD-L1 can vary significantly from one pathology to another. For example, a level of sPD-L1 of 63.9 pg/mL was measured in the serum of patients with acute pancreatitis, significantly higher than the level measured in the control group (48.1 pg/mL) [136]. However, this level is low compared to the level measured in the serum of patients with advanced metastatic gastric cancer (704 pg/mL) [156]. Moreover, in all cases, the level of sPD-L1 is determined by an ELISA method, which does not inform on the biochemical form of the protein (full length, variant). Knowing that there is a heterogeneity of forms, a proper characterization of the sPD-L1 form is strongly recommended to better appreciate its potential function. sPD-L1 is often insufficiently characterized from a biochemical view.

### 4.1. sPD-L1 in Pulmonary Diseases

In idiopathic pulmonary fibrosis (IPF), an overexpression of mPD-L1 has been reported on different cell types, not only invasive lung fibroblasts but also alveolar macrophages, whereas PD-1 was found to be overexpressed on CD^4+^ T-cells, and to contribute to the inhibition of their differentiation into Treg cells [82,84,157]. Targeting the PD-1/PD-L1 checkpoint is considered an option for the treatment of this rare but progressive and fatal lung disease [83]. A high level of circulating sPD-L1 has been measured in the serum of IPF patients, more than three times higher than in the healthy control group [27]. At present, the role of sPD-L1 in IPF is unknown, but it could be a useful marker to follow disease progression and response to treatment. Patients with IPF have a greater risk of the severe COVID-19 upon infection by the severe acute respiratory syndrome coronavirus 2 (SARS-CoV-2) which itself can induce post-inflammatory pulmonary fibrosis [158,159]. Advanced COVID-19 is characterized by a massive, unrestrained immune activation (the so-called “cytokine storm”) and important deficiencies in immune regulatory mechanisms such as T regulatory cells [160]. Therefore, an option to combat the disease is to activate anti-inflammatory cells, such as regulatory Treg cells, macrophages and neutrophils, to modulate the immune response via the secretion of cytokines. Another option to reduce the acute respiratory distress syndrome (ARDS) often associated with COVID-19 could be to activate the PD-1 pathway with sPD-L1. Indeed, a recent study has shown that administration of sPD-L1 in mice with direct ARDS relieved inflammatory lung injury and improved the survival rate. The injected soluble protein was found to decrease the number of lung monocyte-derived macrophages (by inducing their apoptosis) and associated proinflammatory markers. sPD-L1 has revealed a clear protective role in ARDS [140]. It is worth noting here that the levels of sPD-1 were also found to be elevated in both the serum and bronchial alveolar lavage fluid of patients with ARDS. It is equally considered a potential biomarker for ARDS [161]. The same conclusion has been drawn in a study of patients with community-acquired pneumonia. The high level of sPD-L1 measured in these pneumonia patients was correlated with the survival prognosis. sPD-L1 level was positively correlated with the pneumonia severity index [141].

### 4.2. sPD-L1 in Inflammatory and Autoimmune Diseases

Elevated levels of sPD-L1 have been measured in many inflammatory diseases, notably acute pancreatitis (AP), which is a relatively common acute critical disease with multiple causative factors (alcohol, microbial infection, intestinal occlusion and dysbiosis of gut microbiota). A major immunosuppression occurs in the early stage of the disease. The PD-1/PD-L1 checkpoint has been implicated in disease progression and infectious complications [162]. sPD-L1 was found to be significantly upregulated in patients with early AP. The circulating protein is suspected to be involved in the development of immunosuppression in the early stage of the disease [136]. Both serum sPD-1 and sPD-L1, negatively correlated with circulating lymphocyte levels, likely contribute to the immunosuppressive process in acute pancreatitis [137]. A major elevation of the level of circulating sPD-L1 might be the cause or the reflect of a generalized inflammation in AP, just as observed in inflammatory types of pancreatic cancer [163].

Significantly elevated levels of sPD-1 in the sera and synovial fluid of patients with rheumatoid arthritis (RA) have been measured. Moreover, a positive correlation was noted between the levels of serum sPD-1 and the titers of rheumatoid factor and the disease activity score [164]. sPD-1 is believed to play an important role as a mediator of inflammation in both rheumatoid arthritis and psoriatic arthritis [150,164,165]. The level of serum sPD-L1 has been related to systemic inflammation in RA. Moreover, a differential regulation of sPD-L1 has been noted during the early and late RA, possibly reflecting an evolution from acute to chronic inflammation. Smoking tends to limit the expression of sPD-L1 in RA patients [166]. Elevated levels of sPD-L1 have been observed in other autoimmune disorders. The serums of both sPD-1 and sPD-L1 have been found to be significantly higher in patients with systemic lupus erythematosus (SLE) than in healthy controls. These two circulating proteins are likely implicated in the immune dysregulation [153]. In contrast, the level of sPD-L2 was found to be significantly lower in SLE patients than in healthy subjects [167]. Another study reported no significant difference in plasma sPD-L1 concentrations between the SLE patient and control groups [168]. The role of sPD-L1 in SLE is not precisely known at present. The protein level is frequently elevated in autoimmune diseases, although it is the circulating level of PD-1, not PD-L1, which is elevated in autoimmune hepatitis [169]. High levels of sPD-1 have also been observed in patients with immune thrombocytopenia (ITP) [170], whereas another study indicated a decreased level of sPD-1 together with decreased levels of sPD-L1 levels in patients with newly diagnosed ITP [145]. High levels of sPD-L1 have also been recorded in patients with cutaneous systemic sclerosis and, interestingly, the level was positively correlated with the severity of skin sclerosis [154]. It is, therefore, a useful marker to follow the evolution of the disease. A parallel can be drawn with oral lichen planus (OLP) which is a T-cell-mediated chronic inflammatory mucocutaneous disease. mPD-L1, expressed on keratinocytes, plays a role in tolerance induction in the inflamed oral mucosa and skin. It has been reported that the levels of both sPD-1 and sPD-L1 proteins in patients with OLP were significantly higher than those in the control group. A negative correlation was noted between sPD-L1 expression level and CD^4+^ T-cells [146]. However, another recent study pointed out the potential dominant role of PD-L1, not PD-L1, in OLP [171]. Nevertheless, the PD-1/PD-L1 pathway is often compromised in chronic inflammatory condition implicating the oral mucosa or the skin. More work is needed to evaluate the expression and function of sPD-L1 in other inflammatory diseases such as myositis, polymyalgia rheumatica and others.

### 4.3. sPD-L1 and Sepsis

The case of sepsis, a systemic inflammatory disease, is peculiar because there are controverted reports. Kawamoto and coworkers have shown that circulating sPD-L1 levels were significantly higher in sepsis, whereas the level of extracellular vesicles PD-L1 was not elevated in sepsis and in systemic inflammatory response syndrome. This study indicated a positive correlation between sPD-L1 levels and sepsis severity, suggesting that the soluble protein could be implicated in the pathogenesis of the disease [152]. sPD-1 has also been positively correlated with the severity of sepsis [172]. However, in another study, the authors found no differences in levels of soluble PD-1 or PD-L1 between patients with sepsis and controls [173]. In patients with acute liver failure (ALF), the level of sPD-L1 is markedly elevated compared to the healthy control. It was found to be particularly high in ALF patients who developed sepsis or had a poor outcome [135]. More work is needed to better comprehend the role of sPD-L1 in the etiology of sepsis. This is important because anti-PD-L1 antibodies (as well as nanobodies and peptides) are being evaluated for the treatment of sepsis [174,175,176]. On the other side, rare cases of sepsis induced by PD-L1 inhibitors have been recorded [177]. It is thus important to clarify the link between sPD-L1 and sepsis, notably pulmonary sepsis.

### 4.4. sPD-L1 in Virus-Mediated Diseases

The levels of sPD-L1 are generally very high in patients with chronic hepatitis C (CHC), and the increase in the circulating level corresponds to a progression of the disease and possibly the generation of hepatocellular carcinoma [142]. sPD-L1 appears as a useful marker of diseases progression and response to treatment in hepatitis, including in cases of hepatitis B virus (HBV)-related hepatocellular carcinoma [178]. sPD-L1 could represent a biomarker of interest in different viral diseases, in particular in HIV-infected individuals. Significantly higher levels of sPD-L1 have been measured in HIV infected subjects than in uninfected adults, including in patients with an undetectable viral load. Moreover, the levels of sPD-L1 were also high in individuals that do not respond to antiretroviral therapy. The circulating protein may thus be particularly useful to follow the treatment response in HIV patients [138]. In HIV-positive patients, the high level of sPD-L1 likely arises from mPD-L1 expressed on monocyte-derived dendritic cells, which at the same time present an elevated level of MMP-2, susceptible to produce sPD-L1 upon proteolysis [139]. The early expressed HIV-1 Tat protein has the capacity to stimulate the expression of mPD-L1 on monocyte-derived dendritic cells, leading to an abnormal hyper-activation of the immune system via the TLR4 pathway [179,180] and MMP-2 can convert part of mPD-L1 into sPD-L1 [139]. The prominent role of the PD-1/PD-L1 checkpoint in HIV infection has opened novel therapeutic perspectives. Vaccine strategies incorporating DNA sequence encoding sPD-1 or sPD-L1 into a vector that expressed specific HIV-1 proteins have been proposed to modulate the immune-response [181,182]. The specific case of SARS-CoV-2 and COVID-19 has been evoked above, together with the role of sPD-L1 in ARDS (see Paragraph 4.1).

### 4.5. sPD-L1 in Pregnancy

sPD-L1 has been detected in the blood of pregnant women and was found to increase throughout the gestation. Its constitutive production is enhanced by IFN-γ, which is expressed by trophoblast-educated macrophages, during placenta development [183]. An abnormal activation of decidual macrophages leads to alteration of the PD-1/PD-L1 pathway and can affect various aspects of placental development. sPD-L1 increases throughout gestation and returns to a control level post-partum. sPD-L1 plays a role, at least in part, as a suppressor of maternal immunity [29]. It is one of the key molecules implicated in the maternal-fetal immune-tolerance, critical for the fetus [184].

sPD-L1 has been also implicated in endometriosis-associated infertility. Elevated levels of sPD-L1 have been measured in the serum and the peritoneal fluid of endometriosis patients compared to controls. The protein could thus represent a biomarker for endometriosis [144]. In cases of preeclampsia (a pregnancy complication characterized by high blood pressure and damages to organs such as the liver and kidneys), the maternal sPD-L1 level can increase a little but it is essentially sPD-1 levels which are significantly enhanced in preeclamptic women compared to normotensive pregnant women [185].

## 5. Discussion

The soluble, circulating form of PD-L1 has been largely investigated in cancers to determine its value as a biomarker of disease progression, response to checkpoint immunotherapy or treatment resistance. In some cases, sPD-L1 has clearly emerged as a useful biomarker to predict the disease prognosis, such as in peripheral T-cell lymphoma. In this specific case, a high sPD-L1 level was well correlated with a worse clinical response [186]. However, overall, the results are contrasted, and sPD-L1 alone does not appears as a sufficiently robust and reliable biomarker to predict outcome in most cases. In NSCLC, for example, some studies underlined that there is no difference in survival outcomes between low sPD-L1 and high sPD-L1 patients with NSCLC [123], whereas other studies indicated that plasma sPD-L1 (and sPD-L1/sPD-1 ratio) can be positively correlated with overall survival of NSCLC patients [126]. In parallel, other investigations concluded that exoPD-L1, but not sPD-L1, can be correlated with NSCLC disease progression [187]. At present, the significance of sPD-L1 as a prognostic or predictive marker in lung cancer remains uncertain [35]. A composite biomarker including both sPD-1 and sPD-L1 has been proposed to predict nivolumab efficacy in NSCLC patients [188]. The expression levels of sPD-L1 in cancers are extremely variable and not always correlated with the levels of expression of mPD-L1. The variability of sPD-L1 expression may come from at least two factors: the heterogeneity of the biochemical/physiological origin of sPD-L1, and the heterogeneity of the cellular origin and functions of sPD-L1.

A portion of sPD-L1 may also correspond to degradation products of the membrane-bound receptor. mPD-L1 undergoes degradation in proteasomes or lysosomes by multiple pathways, and the mechanisms can be altered in cancer cells [189]. There is an intracellular expression of PD-L1 in recycling endosomes, actively trafficking to the plasma membrane. The degradation is intracellular, through endo-lysosomal, autophagic, proteasomal, or endoplasmic reticulum-related pathways [190], but there may be degradation products deriving from mPD-L1, or more likely from exoPD-L1. The balance among sPD-L1, mPD-L1 and exoD-L1 warrants further investigation.

There is not a unique sPD-L1 species but a pool of circulating proteins, originating from either the proteolytic processing of mPD-L1/exPD-L1, or from the alternative splicing of the *PD-L1* pre-mRNA. Moreover, in both cases, there are several variants: (i) proteolytic variants due to the cleavage of membrane PD-L1 by MMPs, ADAMs and probably other proteases, and (ii) splicing variants due to multiple sites and splice processes. With a variable biochemical origin, it is not surprising that the level and functions of the protein can be distinct from one individual to another, from one disease to another. Moreover, further adding to the complexity of the situation, it has been shown recently that the *N*-glycosylation pattern of the secreted PD-L1 protein is distinct from that of mPD-L1 in cells. In MDA-MB-231 breast cancer cells, full-length mPD-L1 was found to carry mostly complex glycans with a high proportion of poly-*N*-acetyl-lactosamine (poly-LacNAc) structures at the N219 position, whereas sPD-L1 demonstrated a low N219 occupancy with little contribution of the polyLacNAc motif [191]. The different biochemical states determine the heterogenous functionality.

In addition, sPD-L1 can be produced by a variety of cell types, cancer cells primarily but also a diversity of non-malignant cells, including endothelial and epithelial cells. The soluble protein is expressed and secreted in many pathological situations (Table 1) but also in non-pathological circumstances, such as pregnancy. Recently, an increased level of sPD-L1 has been measured in the frame of physical activities. Both moderate-intensity and high-intensity interval exercises produced a marked increase in the plasma level of sPD-L1, with a reciprocal decline in sPD-1 (up to 1 h after exercise). The variations reflect an anti-inflammatory response to exercise [192]. sPD-L1 is more a general marker of an inflammatory status than just a marker of active, immuno-suppressive cancer cells. It reflects the function of mPD-L1 ubiquitously expressed in inflamed tissues. However, sPD-L1 contributes to immune regulation together with mPD-L1 and exPD-L1. Altogether, these different PD-L1 forms reinforce the dynamic crosstalk between the variety of cells implicated in the system. sPD-L1 is an active entity that must be also considered in the design of therapeutic molecules targeting the immune checkpoint. The development of small molecules’ modulators of mPD-L1 is a very active field [193], notably with compounds capable of inducing dimerization or stabilizing PD-L1 dimers [194,195,196,197]. These drug design approaches should consider further the specific value of sPD-L1 as a potential co-target capable of influencing the efficacy of the designed ligands. Clinical approaches aimed at eliminating circulating forms of PD-L1 (plasmatic sPD-L1- and exoPD-L1) by therapeutic plasma exchange are now emerging, and they may be very useful in oncology [198] and possibly in other pathologies. sPD-L1 is implicated in many diseases and conditions. It is a key circulating protein, with both physiological and pathological roles well beyond cancer. The protein is associated with structural heterogeneity leading to a functional diversity. The plurality of sPD-L1 species should be kept in mind when studying this protein.

## 6. Conclusions

Our analysis underlines the diversity of the circulating soluble PD-L1 species, which can be generated by different proteolytic or splicing processes. Elevated levels of sPD-L1 protein have been measured in many types of cancer, and in multiple inflammatory pathologies well beyond cancer, as well as in the frame of pregnancy. sPD-L1 is frequently detected in the plasma or serum, but its exact biochemical nature and functions are insufficiently documented. More work is needed to better characterize the contribution of sPD-L1 to the activity of the PD-L1/PD-1 checkpoint, and to help understanding the dynamic interplay between the different forms of PD-L1, membrane, exosomal, cytoplasmic, nuclear, and soluble. Nevertheless, sPD-L1 is an important multi-faceted circulating protein, associated with a great diversity of diseases. Its contribution to disease progression is not always clear. Hopefully, the review will encourage further studies in this direction.

## Figures and Tables

**Figure 1 cancers-13-03034-f001:**
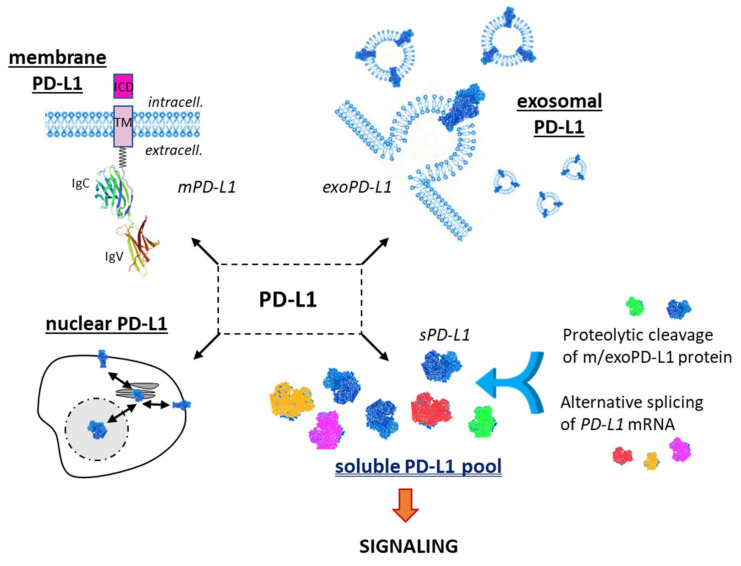
The different forms of the checkpoint protein PD-L1: membrane form (mPD-L1) with its characteristic transmembrane domain (TM) and intracellular domain (ICD); the exosomal form (exoPD-L1) embarked into extracellular vesicles released by a variety of cells; the nuclear form (nPD-L1) involved in the regulation of mRNA stability; the soluble forms (sPD-L1), which can derive either from proteolytic cleavage of m/exoPD-L1 or from an alternative mRNA splicing. m/exo/sPD-L1 can modulate T-lymphocyte activity via the PD-1 signaling pathway.

**Figure 3 cancers-13-03034-f003:**
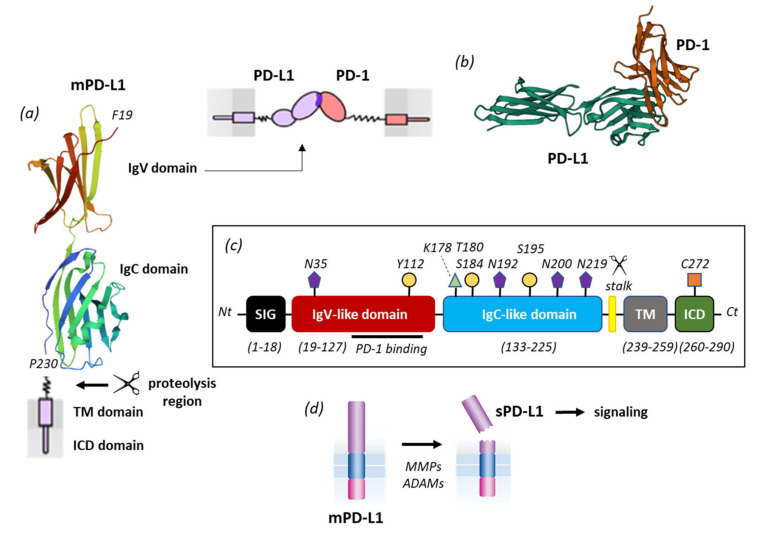
PD-L1 primary structure and processing. (**a**) mPD-L1 with its IgV domain (which interacts with PD-1) and IgC domain linked to the transmembrane domain (TM) via a short flexible stalk. The TM domain is connected to the intracellular (ICD) domain. The molecular model of PD-L1 derives from the crystal structure of the free protein (PDB access code 3BIS). (**b**) Model of PD-1 interacting with PD-L1 via its IgV domain. (**c**) Primary structure of mPD-L1 with the different domains and sites of post-translational modifications (*N*-glycosylation at N35, N192, N200 and N219; phosphorylation at Y112, T180, S184, S195, ubiquitination at K178 and palmitoylation at C272). Proteolytic cleavage generally occurs within a short stalk region situated between the IgC and TM domains. (**d**) Representation of mPD-L1 shedding by different proteases (MMPs, ADAMs) to release a soluble form of PD-L1, which plays a role in cell signaling.

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
