# Peer review of "Soluble Programmed Death Ligand-1 (sPD-L1): A Pool of Circulating Proteins Implicated in Health and Diseases"

_cancers, 2021, doi:10.3390/cancers13123034_

Round 1

Reviewer 1 Report

The review “Soluble programmed death ligand-1 (sPD-L1): a pool of circulating proteins implicated in health and diseases” by Bailley et al summarizes evidence for the functional importance of sPD-L1. It is a quite comprehensive review but the review lacks a critical evaluation of the results. In the present from, it provides a cumulative list of papers without a discussion of the strength of the provided evidence. The readers therefore cannot understand whether the evidence is as impressive as the number of cited papers might indicate. The comments below suggest which issues need a critical review and would benefit from further discussion.

Major issues:

  1. The premise of the paper is that “soluble” PD-L1 is physiologically relevant, i.e. detectable in vivo and functionally important. The “soluble” seems to include PD-L1 in exosomes, is that correct? Please define it clearly and if not, why the extensive discussion of exosomal PD-L1? Is that because it is hardly ever distinguished from the shed form? If yes, the exosomal form represents a membrane bound full length form so why would it be considered soluble? The biggest problem is that the evidence for the sPD-L1 (its presence or function) could be attributed to the exosomal form in many of the cited papers. For example, evidence for sPD-L1 in circulation (plasma/serum) comes from ELISA measurements that will not distinguish between the vesicular of soluble forms. The authors need to discuss this critically and state which papers (if any) went through the effort to distinguish the forms.
  2. The evidence for sPD-L1 detectable in plasma/serum (in circulation in vivo) comes from ELISA tests. I did not see any evidence, in any paper cited, that the form of the sPD-L1 was determined (by any alternative method). Yet, the references cite ng/ml amounts of sPD-L1; this amount could be enriched/analyzed relatively easily. Ref 86 apparently detected glycosylated/deglycosylated PD-L1 in plasma (Figure 4a therein) but this evidence is not convincing or sufficient (and is the best evidence I found). The authors need to provide a critical assessment of the evidence and document whether the proteins were fully verified as the long form of sPD-L1.
  3. The cleaved PD-L1 is suggested to derive from proteolytic cleavage in the short linker region (see Figure 3 legend and Figure 4); the authors claim that such cleavage by MMPs and ADAMs releases a nearly full length functional protein able to bind to PD-1. While some evidence in the cell lines may be supportive of this fact, I did not see any paper that documents the cleavage in this region. Why is that? The authors need to critically discuss which papers provide clear evidence of cleavage in this region and which papers fully document sPD-L1 in full length form.

This means that, as opposed to release of a long functional protein, many of the papers may be describing degradation of the membrane bound receptor as a cause of the functional effect. The loss of membrane form (to proteolytic degradation or other means) does on equate to sPD-L1 release. That needs to be clearly distinguished in the discussion These issues are mixed up in the review in a way that is likely to confuse the readers.

  1. The authors state on page 8 line 313 “We will not comment further the biomarker aspect of sPD-L1, there are specific recent reviews on this topic”. That is OK but including nearly 25 biomarker references in support of PD-L1 without discussing quality of the evidence is not. Quantitative aspects of the evidence are not discussed. Please provide a critical comparison of the following: a. amounts of sPD-L1 detected in the cited studies; b. quality of the evidence; c. form of the protein detected (exosomal or soluble and what size/form). A comparison of the ELISA assays, or other methods of detection, will be useful and can be done instead of the repetition of the already reviewed biomarker results.
  2. In view of the above, I consider Figure 2 misleading. Please provide cited references for each case of disease where sPD-L1 was unequivocally detected. Table 1 provides refs for non-malignant disease but what did the references really measure? If ELISA, how do you know they measured sPD-L1 or that it has any functional implications? Are there any papers where sPD-L1 was verified and documented as functionally important? If so, please list them and provide a critical review compared to the others.

In fact, sPD-L1 is detected in healthy controls and very often at comparable quantities to the disease groups. So why would it matter in all the diseases listed in Figure 2 or Table 1?

  1. sPD-L1 can function in vivo only if its concentration is sufficient. This is not my expertise but therapeutic antibodies are administered at mg/kg quantities and achieve microgram/ml plasma concentrations (for weeks) while sPD-L1 is in ng/ml quantities even if we believe the existing data (see above). Is that enough? What is the evidence that the sPD-L1 is high enough at the site of action? And does the immunological synapse matter? The synapse is not simply sPD-L1/PD-1 but multiple interactions that depend on the membrane presentation. Please provide a critical discussion.
  2. Figure 1 lists membrane/vesicular/soluble forms. I do not understand from the Scheme/paper whether exosomal (vesicular) is considered soluble (see above).

In addition, how about intracellular forms (PMID: 31053471 etc)? The membrane bound PD-L1 is clearly recycled and further processed (without shedding); the recycling is entirely ignored in this review. That is fine, the review can focus on the sPD-L1, but why include the membrane/vesicular forms and exclude intracellular forms in Figure 1? It would be better to include it and state the reasons why it is not discussed.

Author Response

Comments and Suggestions for Authors

The review “Soluble programmed death ligand-1 (sPD-L1): a pool of circulating proteins implicated in health and diseases” by Bailley et al summarizes evidence for the functional importance of sPD-L1. It is a quite comprehensive review but the review lacks a critical evaluation of the results. In the present from, it provides a cumulative list of papers without a discussion of the strength of the provided evidence. The readers therefore cannot understand whether the evidence is as impressive as the number of cited papers might indicate. The comments below suggest which issues need a critical review and would benefit from further discussion.

Major issues:

1. The premise of the paper is that “soluble” PD-L1 is physiologically relevant, i.e. detectable in vivo and functionally important. The “soluble” seems to include PD-L1 in exosomes, is that correct? Please define it clearly and if not, why the extensive discussion of exosomal PD-L1? Is that because it is hardly ever distinguished from the shed form? If yes, the exosomal form represents a membrane bound full length form so why would it be considered soluble? The biggest problem is that the evidence for the sPD-L1 (its presence or function) could be attributed to the exosomal form in many of the cited papers. For example, evidence for sPD-L1 in circulation (plasma/serum) comes from ELISA measurements that will not distinguish between the vesicular of soluble forms. The authors need to discuss this critically and state which papers (if any) went through the effort to distinguish the forms.

  • This is an excellent point which we had neglected. Thanks to the reviewer, we have now added a specific paragraph about this important issue (page 4, lines 152-160).

2. The evidence for sPD-L1 detectable in plasma/serum (in circulation in vivo) comes from ELISA tests. I did not see any evidence, in any paper cited, that the form of the sPD-L1 was determined (by any alternative method). Yet, the references cite ng/ml amounts of sPD-L1; this amount could be enriched/analyzed relatively easily. Ref 86 apparently detected glycosylated/deglycosylated PD-L1 in plasma (Figure 4a therein) but this evidence is not convincing or sufficient (and is the best evidence I found). The authors need to provide a critical assessment of the evidence and document whether the proteins were fully verified as the long form of sPD-L1.

  • We agree with this remark. A comment has been added page 9 (lines 351-354) and page 12 (lines 454-463). sPD-L1 is often insufficiently characterized.

3. The cleaved PD-L1 is suggested to derive from proteolytic cleavage in the short linker region (see Figure 3 legend and Figure 4); the authors claim that such cleavage by MMPs and ADAMs releases a nearly full-length functional protein able to bind to PD-1. While some evidence in the cell lines may be supportive of this fact, I did not see any paper that documents the cleavage in this region. Why is that? The authors need to critically discuss which papers provide clear evidence of cleavage in this region and which papers fully document sPD-L1 in full length form.

  • OK, there is no definitive evidence that that the cleavage occurs in this region. But this idea has been proposed by Romero et al. [50] and we clearly indicated that this is a hypothesis (The position of the mPD-L1 cleavage site by ADAMs is not precisely known but the authors postulated that it is located in the stalk region, between V225 and H240, based on the analysis of the size of the degradation products (Figure 4) [50].”).

We have modified the text page 7 (line 318) and the Fig. 4 to underline this point. The hypothesis makes sense, we would like to maintain it.

This means that, as opposed to release of a long functional protein, many of the papers may be describing degradation of the membrane bound receptor as a cause of the functional effect. The loss of membrane form (to proteolytic degradation or other means) does on equate to sPD-L1 release. That needs to be clearly distinguished in the discussion These issues are mixed up in the review in a way that is likely to confuse the readers.

  • We agree. OK, we did not sufficiently consider the possibility that part of sPD-L1 can result from a degradation of the exoPD-L1 as a membrane receptor. The discussion has been amended (page 15, lines 623-632) to raise this point.

4. The authors state on page 8 line 313 “We will not comment further the biomarker aspect of sPD-L1, there are specific recent reviews on this topic”. That is OK but including nearly 25 biomarker references in support of PD-L1 without discussing quality of the evidence is not. Quantitative aspects of the evidence are not discussed. Please provide a critical comparison of the following: a. amounts of sPD-L1 detected in the cited studies; b. quality of the evidence; c. form of the protein detected (exosomal or soluble and what size/form). A comparison of the ELISA assays, or other methods of detection, will be useful and can be done instead of the repetition of the already reviewed biomarker results.

  • An extensive quantitative comparison is not possible. The reported numbers for sPD-L1 vary considerably from one study to another. Even in a more or less homogenous situation (NSCLC), the reported sPD-L1 serum levels are extremely variable. We have added a comment (page 9, lines 463-474) to underline this aspect.

5. In view of the above, I consider Figure 2 misleading. Please provide cited references for each case of disease where sPD-L1 was unequivocally detected. Table 1 provides refs for non-malignant disease but what did the references really measure? If ELISA, how do you know they measured sPD-L1 or that it has any functional implications? Are there any papers where sPD-L1 was verified and documented as functionally important? If so, please list them and provide a critical review compared to the others.

  • We do not consider that Fig. 2 is misleading; the legend is clear, it is a “non-exhaustive list”, useful to illustrate the topic (sPD-L1 is multiple associated with diseases, beyond cancer). We would like to maintain this Figure which nicely illustrates the review.
  • We agree with the fact that sPD-L1 is certainly not always sufficient characterized. The new Conclusion refers to this point.

In fact, sPD-L1 is detected in healthy controls and very often at comparable quantities to the disease groups. So why would it matter in all the diseases listed in Figure 2 or Table 1?

  • We do not agree with this comment. Yes, sPD-L1 is detected in healthy controls but Not “very often at comparable quantities to the disease groups”. Many studies point to an elevated levels of sPD-L1 in patients, such as in the case of the diseases indicated in Table 2. This is the interest of this specific Table.

6. sPD-L1 can function in vivo only if its concentration is sufficient. This is not my expertise but therapeutic antibodies are administered at mg/kg quantities and achieve microgram/ml plasma concentrations (for weeks) while sPD-L1 is in ng/ml quantities even if we believe the existing data (see above). Is that enough? What is the evidence that the sPD-L1 is high enough at the site of action? And does the immunological synapse matter? The synapse is not simply sPD-L1/PD-1 but multiple interactions that depend on the membrane presentation. Please provide a critical discussion.

  • We cannot compare with humanized antibodies, which are subject to an intense recirculation (via FcRn recognition) and administered at a high dose. We agree that sPD-L1 is present in small quantities and that its functionality needs to be better defined. This is clearly indicated.

7. Figure 1 lists membrane/vesicular/soluble forms. I do not understand from the Scheme/paper whether exosomal (vesicular) is considered soluble (see above).

  • We considered exosomal (vesicular) PD-L1 as a distinct form from sPD-L1.

In addition, how about intracellular forms (PMID: 31053471 etc)? The membrane bound PD-L1 is clearly recycled and further processed (without shedding); the recycling is entirely ignored in this review. That is fine, the review can focus on the sPD-L1, but why include the membrane/vesicular forms and exclude intracellular forms in Figure 1? It would be better to include it and state the reasons why it is not discussed.

  • OK, we agree. The recycling of the receptor is now evoked in the Discussion (with appropriate refences) and the intracellular forms are cited in the Introduction, with references indicated (PMID: 31053471 and two other recent works). Fig. 1 has been reworked to mention nPD-L1. As indicated, we do not discuss the intracellular PD-L1 protein further, for the sake of clarity.

Altogether, we made substantial changes to take into account all comments of the reviewer. The modifications have contributed to improve the Ms. Thank you.

Reviewer 2 Report

  1. Authors discussed mechanisms in terms of generation of soluble PD-L1: proteolysis and alternative splicing. Intriguing questions regarding how the process of proteolytic cleavage and/or alternative splicing is controlled, and how the balance among sPD-L1, mPD-L1 and exoD-L1 is maintained in cells, particularly cancer cells, should be discussed.
  2. To more appropriately fit the scope of the journal - Cancers, it is highly recommended for authors to expand their description of mPD-L1 on cancer cells for a broader group of readers in the field, in particular CTCs (Kloten et al., 2019 Cells 8:809) and CTECs (Zhang et al., 2020 Cancer Letters 469:355).

Author Response

  1. Authors discussed mechanisms in terms of generation of soluble PD-L1: proteolysis and alternative splicing. Intriguing questions regarding how the process of proteolytic cleavage and/or alternative splicing is controlled, and how the balance among sPD-L1, mPD-L1 and exoD-L1 is maintained in cells, particularly cancer cells, should be discussed.
  • The regulation of the equilibrium between the different forms of PD-L1 is not known, but this is a good point. A new paragraph in the Discussion (page 15, lines 623-630) addresses this issue.

2. To more appropriately fit the scope of the journal - Cancers, it is highly recommended for authors to expand their description of mPD-L1 on cancer cells for a broader group of readers in the field, in particular CTCs (Kloten et al., 2019 Cells 8:809) and CTECs (Zhang et al., 2020 Cancer Letters 469:355).

  • The two references have been added (lines 103-107) and a comment about CTC measurements incorporated.

We thank the reviewer for the helpful and constructive comments.

Reviewer 3 Report

The article has some issues with consistency. The references are relevant and recent. The cited sources are referenced correctly. Appropriate and key studies are included. The paper lacks a comprehensive view; the flow loses logic at some points. However, the data is presented critically.

The quality of figures and tables is high.
There are some specific comments on weaknesses of the article and what could be improved:
Specific comments on weaknesses of the article and what could be improved:
Major points - the text should be divided into more paragraphs, and not just written without links between the different passages and sections.
Minor points

  1. The abstract should be shortened (i.e., the generation of isoforms)
  2. Consider using subtitles in the first section The PD-1/PD-L1 checkpoint, as well, to make the text more clear and comprehensive
  3. Line 433 "A major immunosuppression occurs at early stage of the disease which can be fatal." Please, clarify.
  4. The conclusion is relatively long but not concise.

Author Response

The article has some issues with consistency. The references are relevant and recent. The cited sources are referenced correctly. Appropriate and key studies are included. The paper lacks a comprehensive view; the flow loses logic at some points. However, the data is presented critically.

The quality of figures and tables is high.
There are some specific comments on weaknesses of the article and what could be improved:
Specific comments on weaknesses of the article and what could be improved:
Major points - the text should be divided into more paragraphs, and not just written without links between the different passages and sections.

  • OK, new sub-paragraphs have been added (1.1, 1.2, 1.3, 3.1, 3.2)

Minor points

  1. The abstract should be shortened (i.e., the generation of isoforms)
  • OK, we have slightly shortened the abstract (269 words instead of 283 words initially).

2. Consider using subtitles in the first section The PD-1/PD-L1 checkpoint, as well, to make the text more clear and comprehensive

  • OK, subtitles have been added (1.1, 1.2, 1.3, 3.1, 3.2).

3. Line 433 "A major immunosuppression occurs at early stage of the disease which can be fatal." Please, clarify.

  • The term “which can be fatal" has been deleted. It was not clear indeed.

4. The conclusion is relatively long but not concise.

  • The “Conclusion” has been renamed “Discussion”. It is a little longer because we have integrated the comment of reviewer 1 about PD-L1 trafficking and intracellular forms.
  • We have introduced a short new section “Conclusion”, to sum up the main message of the review and to mention the limitation associated with sPD-L1.

We thank the reviewer for the helpful and constructive comments.

Round 2

Reviewer 1 Report

Dear authors, thank you for your updates. I have one last concern and that is Figure 2. Please provide references directly in the table supporting the claim that the sPD-L1 is indeed meaningfully increased in that context; please remove all other diseases listed. In my opinion, the Figure is an very optimistic overstatement but I may have missed the supporting literature.  

Author Response

Dear reviewer. Thanks for the comment. We would like to maintain Fig. 2 which we consider useful and attractive. The Figure has been slightly modified (there was a duplicated term). The legend has been completed to cite the studies that pertain to the enhanced expression of sPD-L1 in each cancer type and/or its pronostic value. 22 new references have been incorporated. I hope that this change will adress adequately your remark. Thanks again for your analysis.

Reviewer 3 Report

The authors improved significantly their work by implementing all the suggestions of the reviewers. 

Author Response

OK, thank you.